# Severe leptospirosis in tropical Australia: Optimising intensive care unit management to reduce mortality

Simon Smith[1], Yu-Hsuan Liu[2], Angus Carter[2,3], Brendan J. Kennedy[4,5], Alexis Dermedgoglou[1], Suzanne S. Poulgrain[1], Matthew P. Paavola[1], Tarryn L. Minto[1], Michael Luc[1], Josh Hanson[1,6]*

1 Department of Medicine, Cairns Hospital, Cairns, Queensland, Australia, 2 Department of Intensive Care, Cairns Hospital, Cairns, Queensland, Australia, 3 James Cook University, Cairns Campus, Cairns, Queensland, Australia, 4 Infectious Diseases Service, Royal Adelaide Hospital, Adelaide, South Australia, Australia, 5 Communicable Disease Control Branch, Adelaide, South Australia, Australia, 6 The Kirby Institute, University of New South Wales, Sydney, Australia

* jhanson@kirby.unsw.edu.au

**Data Availability Statement:** Data cannot be shared publicly because of the Queensland Public Health Act 2005. Data are available from the Far

## Abstract

### Background

Severe leptospirosis can have a case-fatality rate of over 50%, even with intensive care unit (ICU) support. Multiple strategies–including protective ventilation and early renal replacement therapy (RRT)–have been recommended to improve outcomes. However, management guidelines vary widely around the world and there is no consensus on the optimal approach.

### Methodology/Principal findings

All cases of leptospirosis admitted to the ICU of Cairns Hospital in tropical Australia between 1998 and 2018 were retrospectively reviewed. The patients' demographics, presentation, management and clinical course were examined. The 55 patients' median (interquartile range (IQR)) age was 47 (32–62) years and their median (IQR) APACHE III score was 67 (48–105). All 55 received appropriate antibiotic therapy, 45 (82%) within the first 6 hours. Acute kidney injury was present in 48/55 (87%), 18/55 (33%) required RRT, although this was usually not administered until traditional criteria for initiation were met. Moderate to severe acute respiratory distress syndrome developed in 37/55 (67%), 32/55 (58%) had pulmonary haemorrhage, and mechanical ventilation was required in 27/55 (49%). Vasopressor support was necessary in 34/55 (62%). Corticosteroids were prescribed in 20/55 (36%). The median (IQR) fluid balance in the initial three days of ICU care was +1493 (175–3567) ml. Only 2/55 (4%) died, both were elderly men with multiple comorbidities.

### Conclusion

In patients with severe leptospirosis in tropical Australia, prompt ICU support that includes early antibiotics, protective ventilation strategies, conservative fluid resuscitation, traditional

North Queensland Human Research Ethics Committee (contact via email Cairns_Ethics@health.qld.gov.au) for researchers who meet the criteria for access to confidential data.

**Funding:** The authors received no specific funding for this work.

**Competing interests:** The authors have declared that no competing interests exist.

thresholds for RRT initiation and corticosteroid therapy is associated with a very low case-fatality rate. Prospective studies are required to establish the relative contributions of each of these interventions to optimal patient outcomes.

## Author summary

Severe leptospirosis continues to carry a high mortality. To improve outcomes, several countries have developed national guidelines for the management of severe disease. However, there is substantial variation in these guidelines despite the fact that severe leptospirosis has a very similar presentation around the world. In tropical Australia, the case-fatality rate of severe leptospirosis is only 4%. This retrospective study reviewed 55 patients with leptospirosis requiring support in an Australian Intensive Care Unit (ICU) in an effort to identify the management strategies that might explain these excellent outcomes. The low case-fatality rate was associated with prompt multimodal ICU support that included early antibiotics, protective ventilation strategies, conservative fluid resuscitation, traditional thresholds for renal replacement therapy initiation and corticosteroid therapy. However prospective, multinational studies–which include the resource-poor settings that bear the greatest burden of disease–are necessary to define which of these interventions have the greatest therapeutic value.

## Introduction

Leptospirosis, a zoonotic disease with a global distribution, is estimated to kill 60,000 people annually [1]. Although most infections are mild, severe disease–manifesting as acute kidney injury (AKI), pulmonary haemorrhage, acute respiratory distress syndrome (ARDS) and multiorgan failure–occurs in up to 15% of cases. Even with intensive care unit (ICU) support, the case-fatality rate of severe leptospirosis can rise to 52% [2].

Multiple, country-specific guidelines have been developed to help improve the management of the critically ill patients suffering from the disease [3–5]. However, although the clinical presentation of severe leptospirosis is similar around the world [6], there is significant variation in the recommendations provided in different countries' guidelines.

Pulmonary involvement in leptospirosis–which manifests as intra-alveolar haemorrhage and ARDS–carries a particularly poor prognosis [7]. It is notable that protective lung ventilation strategies were first developed to manage patients with leptospirosis to prevent the alveolar collapse and over distension seen in the disease [8]. However, the role of non-invasive ventilation and the optimal timing and indications for intubation and mechanical ventilation differ substantially between guidelines [3–5].

Renal replacement therapy (RRT) improves outcomes in patients with leptospirosis-related AKI [9, 10]. Early haemodialysis using criteria (urea >20 mmol/L, serum potassium >4 mmol/L and oligo-anuria without hypovolemia) that promote aggressive use of RRT, was associated with a mortality of only 6% in an ICU cohort from Réunion Island [10]. However, these suggested thresholds are quite different to traditional criteria for RRT initiation which include severe hyperkalaemia, severe acidosis, refractory fluid overload and uraemic complications [11]. The thresholds proposed by the Réunion Island investigators therefore require external validation, particularly as RRT is invasive, expensive and–in many countries–difficult to access [12].

Fluid management in severe leptospirosis is challenging, particularly in the setting of AKI, myocarditis or the multiple organ dysfunction syndrome. A conservative approach may be more appropriate given the high mortality associated with pulmonary involvement [13]. However some national guidelines recommend aggressive fluid replacement with diuretics in those with oliguric renal failure and hypovolaemia [5].

Finally, while antibiotics are universally recommended for severe leptospirosis [3–5], the role of corticosteroids in pulmonary leptospirosis is undefined [14]. Some studies have suggested benefit [15, 16], but an increase in nosocomial infections has also been described [17]. Nonetheless, the early initiation of intravenous corticosteroids is recommended in some national guidelines [3, 5].

In Queensland, Australia, the incidence of leptospirosis is one of the highest in the developed world [18]. Among 402 adults with laboratory-confirmed leptospirosis in Far North Queensland, Australia over a 19-year period, 50 (12%) had severe disease, 45 (90%) of whom required ICU admission [6]. Although the patients' APACHE III scores were high (median: 84 (range 27–169)), only 2/50 (4%) died. This study was performed to examine the different interventions employed in this region, in an effort to inform the optimal approach to the critically ill patient.

## Methods

### Study population and data collection

This retrospective study was performed at Cairns Hospital, a 531-bed tertiary hospital in tropical Australia. The hospital has the region's sole ICU and serves a population of 280,000 people, dispersed over an area of 380,000 km$^2$. We examined all microbiologically confirmed cases of leptospirosis admitted to the Cairns Hospital ICU between January 1998 and July 2018. Microbiological confirmation required one or more of the following criteria: (1) Microscopic agglutination test (MAT) single titre of $\geq$1:400; (2) Fourfold rise in MAT antibody titres; (3) Leptospires isolated from blood culture; (4) Detection of Leptospira in blood by polymerase chain reaction (PCR).

The medical record of each case was reviewed and the following data collected: patient demographics, active comorbidities (defined as requiring current medication or requiring ongoing medical review), clinical features at hospital presentation and on admission to the ICU, haematological and biochemical investigations, imaging results, indication for ICU admission, duration of ICU stay and outcome. AKI was defined using the Kidney Disease: Improving Global Outcomes (KDIGO) guidelines (an increase in serum creatinine $\geq$0.3mg/dL ($\geq$26.5 μmol/L) or increase in serum creatinine to $\geq$1.5 times baseline) [12]; ARDS was defined using the Berlin Definition (mild (200 mm Hg < PaO2/FIO2 $\leq$ 300 mm Hg), moderate (100 mm Hg < PaO2/FIO2 $\leq$ 200 mm Hg), and severe (PaO2/FIO2 $\leq$ 100 mm Hg)) [19] and pulmonary haemorrhage was defined as the presence of frank haemoptysis or gross blood on tracheal aspirate. The patient's care in the ICU was also reviewed with data collected on the duration, type and timing of antibiotic therapy, mechanical ventilation, RRT, vasopressor support and fluid management and whether or not corticosteroids were administered. The Simplified Acute Physiology Score (SAPS) II, Sequential Organ Failure Assessment (SOFA) and Acute Physiology And Chronic Health Evaluation (APACHE) III scores were calculated in each case [20–22]. A Murray score was calculated in intubated patients, where possible, to determine patients' eligibility for extra-corporeal membrane oxygenation (ECMO) [23].

### Ethical considerations

The Far North Queensland Human Research Ethics Committee provided approval for the study (HREC/16/QCH/37 – 1043LR). As the data were retrospective and de-identified, the committee waived the requirement for patient consent.

### Statistical analysis

Data were entered into an electronic database (Microsoft Excel) and analysed using statistical software (Stata 14.2). Groups were analysed using the Kruskal-Wallis, chi-squared and Fisher's exact tests where appropriate.

## Results

There were 55 patients admitted to the ICU with leptospirosis over the 21-year study period. Their median (interquartile range (IQR)) age was 47 (32–62) years; 49/55 (89%) were male. Most cases (35/55 (64%)) occurred during the region's December-April wet season and 44/55 (80%) had an identifiable occupational or recreational exposure to leptospirosis. Over half (28/55 (51%)) initially presented to a regional hospital and required medical retrieval for tertiary care at Cairns Hospital. Only 9/55 (16%) had a documented comorbidity. The patients' demographics and their presenting symptoms are presented in Table 1.

Leptospirosis was the suspected diagnosis in 34/55 (62%) on admission to the ICU, with a different infection being suspected in 17/55 (31%). Leptospirosis was confirmed by PCR in 29/

**Table 1. Baseline characteristics of patients at time of presentation to hospital who subsequently required intensive care unit support for severe leptospirosis in Far North Queensland, Australia between 1997 and 2018.**

| Clinical feature | n (%) |
|---|---|
| Age (years) (median (IQR)) | 47 (32–62) |
| Male | 49 (89) |
| Exposure to leptospirosis | 44 (80) |
| Occupational | 29 (53) |
| Recreational | 17 (31) |
| Any comorbidity | 9 (16) |
| Diabetes mellitus | 4 (7) |
| Cardiovascular disease | 3 (5) |
| Malignancy | 3 (5) |
| Chronic kidney disease | 2 (4) |
| Connective tissue disease | 1 (2) |
| HIV | 1 (2) |
| Duration of symptoms (days) (median (IQR)) | 5 (4–5) |
| Fever ($\geq 38.0\,^\circ$C) | 53 (96) |
| Myalgia or arthralgia | 46 (84) |
| Nausea or vomiting | 45 (82) |
| Headache | 39 (71) |
| Oliguria (urine output $\leq$ 0.5L/24 hours) | 35 (64) |
| Cough | 21 (38) |
| Abnormal bleeding | 21 (38) |
| Abdominal pain | 20 (36) |
| Dyspnoea | 19 (35) |
| Conjunctival suffusion | 16 (29) |
| Clinical jaundice | 4 (7) |

IQR interquartile range; HIV: human immunodeficiency virus

**Table 2. Infecting serovar causing leptospirosis requiring ICU care in Far North Queensland, Australia between 1997 and 2018.**

| Serovar | n (%) |
|---|---|
| Australis | 17 (40) |
| Zanoni | 16 (38) |
| Kremastos | 3 (7) |
| Robinsoni | 2 (5) |
| Hardjo | 1 (2) |
| Mankarso | 1 (2) |
| Szwajizak | 1 (2) |
| Arborea | 1 (2) |

55 (53%), a fourfold rise in MAT antibody titres in 28/55 (51%), blood culture in 16/55 (29%) and by a single MAT titre of $\geq$ 1:400 in 2/55 (4%). The infecting serovar was established in 42/55 (76%) (Table 2). On admission to the ICU, most patients were acidotic, and thrombocytopenia was common, but severe hyperbilirubinaemia was rare (Table 3).

Patients had severe disease: their median (IQR) APACHE III score was 67 (48–105), acute kidney injury and pulmonary involvement were both common, and most patients required some form of organ support (Table 4). Despite this only 2/55 (4%) died. One was a 73-year-old male farmer with known chronic obstructive airways disease and ischaemic heart disease, who had a 4-day history of fevers, nausea, vomiting and breathlessness. He initially presented to a regional hospital with abdominal pain, hypotension and anuria but was quickly transferred by helicopter to Cairns Hospital, where he was found to have an acute kidney injury and severe metabolic acidosis. He was intubated, ventilated and required noradrenaline and vasopressin to maintain a mean arterial pressure (MAP) of 70 mmHg. Antibiotics, intravenous corticosteroids and continuous veno-venous haemofiltration, (CVVHF) were administered promptly, however despite this he died within 24 hours of his initial presentation to hospital. The other death occurred in an 80-year-old man with type 2 diabetes mellitus and treatment-resistant hypertension. An avid gardener, he presented to a regional hospital with 5 days of fevers, lethargy, anorexia, nausea and diarrhoea and on initial review was confused, hypotensive and anuric. Blood tests demonstrated an AKI and severe thrombocytopenia. He was urgently transferred to Cairns Hospital where he became hypoxic and developed severe metabolic acidosis. CVVHF was initiated; he was intubated, ventilated and required adrenaline, noradrenaline and vasopressin to maintain a MAP of 70 mmHg, but he deteriorated and died 40 hours after his initial hospital presentation.

Continuous RRT (CVVHF, continuous veno-venous haemodialysis or continuous venovenous haemodiafiltration) was required in 18/55 (33%). For those requiring RRT, the median (IQR) time to ICU admission was 5.2 (2.8–11.6) hours and their median (IQR) time to RRT was 10.5 (3.4–14.6) hours. The indications for starting RRT were severe, refractory metabolic acidosis (pH <7.2) in 13/18 (72%) or hypoxic respiratory failure with fluid overload and anuric renal failure (4/18 (22%)). Only one patient had not met conventional criteria for RRT at the time of its initiation–he had a pH of 7.23 and anuric renal failure–but given his clinical trajectory, the decision was made to commence CVVHF (Table 5). Of the 18 patients requiring RRT, 16 (89%) also had moderate-severe ARDS and 14 (78%) required intubation and mechanical ventilation.

A total of 13 patients received non-invasive ventilation. Of these, 11/13 (85%) subsequently required intubation and mechanical ventilation. The median (IQR) duration of non-invasive ventilation prior to intubation was 6.5 (2–8.5) hours. Intubation and mechanical ventilation

**Table 3. Investigation parameters of patients with severe leptospirosis at the time of presentation to the Intensive Care Unit in Far North Queensland, Australia between 1997 and 2018.**

| Investigation | Number (total = 55) [a] | Value |
|---|---|---|
| pH | 52 | 7.33 (7.21–7.40) |
| PaCO$_2$ (mmHg) | 52 | 34 (31–46) |
| PaO$_2$/FiO$_2$ ratio (all patients) | 52 | 167 (110–268) |
| PaO$_2$/FiO$_2$ ratio (intubated patients) | 27 | 144 (82–205) |
| White cell count (x 10$^9$/L) | 55 | 9.9 (7.4–17.2) |
| Haemoglobin (g/dL) | 55 | 11.2 (10.1–12.1) |
| Platelets (x 10$^9$/L) | 55 | 74 (45–109) |
| Sodium (mmol/L) | 55 | 135 (132–136) |
| Potassium (mmol/L) | 55 | 3.7 (3.4–4.2) |
| Urea (mmol/L) | 55 | 13.2 (8.4–23.1) |
| Creatinine (μmol/L) | 55 | 240 (146–466) |
| Bilirubin (μmol/L) | 55 | 23 (17–38) |
| Prothrombin time (s) | 55 | 14 (13–15) |
| Creatine kinase (IU/L) | 40 | 344 (106–859) |
| C-reactive protein (mg/L) | 23 | 258 (228–322) |
| Troponin I > 0.35 μg/L (n (%)) | 21 | 11 (52%) |
| Base deficit > 5 mmol/L (n (%)) | 52 | 38 (73%) |
| Lactate > 2.5 mmol/L (n (%)) | 49 | 11 (22%) |

All numbers represent median (interquartile range) or n (%)

[a] Not all investigations were performed in every patient

were required in 27/55 (49%) patients (Table 6), 7 (26%) of whom required intubation at a regional hospital prior to transfer to Cairns Hospital. Prone ventilation was employed in 4/27

**Table 4. Manifestations of severe leptospirosis and need for organ support in Far North Queensland, Australia between 1997 and 2018.**

| Characteristic | n (%) |
|---|---|
| Acute kidney injury | 48 (87) |
| Oliguria (urine output ≤ 0.5L/24 hours) | 35 (64) |
| Moderate-severe ARDS | 37 (67) |
| Acute kidney injury and moderate-severe ARDS | 35 (64) |
| Shock [a] | 34 (62) |
| Intra-alveolar haemorrhage | 32 (58) |
| SOFA score (median (IQR)) | 11 (7–17) |
| SAPS II score (median (IQR)) | 32 (17–55) |
| APACHE III score (median (IQR)) | 67 (48–105) |
| Renal replacement therapy | 18 (33) |
| Non-invasive ventilation | 13 (24) |
| Intubation and mechanical ventilation | 27 (49) |

Abbreviations: ARDS, acute respiratory distress syndrome; SOFA, Sequential Organ Failure Assessment; IQR, interquartile range; SAPS, Simplified Acute Physiology Score; APACHE, Acute Physiology Age Chronic Health Evaluation. Oliguria: Urine output ≤ 0.5L/24 hours

[a] defined as blood pressure < 90/60 mmHg despite appropriate intravenous fluid challenge and requiring vasopressor support

**Table 5. Renal investigations of patients with severe leptospirosis requiring renal replacement therapy in Far North Queensland, Australia between 1997 and 2018.**

| Investigation | At time of ICU admission Median (IQR) | At time of RRT initiation Median (IQR) |
|---|---|---|
| Urea (mmol/L) | 23.1 (19.9–29.4) | 25.9 (20.7–29.5) |
| pH | 7.23 (7–7.34) | 7.14 (7.04–7.24) |
| Base deficit (mmol/L) | 10.5 (7.6–14.8) | 14.0 (9.0–17.0) |
| Potassium (mmol/L) | 4 (3.6–4.6) | 4.4 (3.8–5.0) |

Abbreviations: ICU, intensive care unit; IQR, interquartile range; RRT, renal replacement therapy

(15%) and prostacyclin was used in 6/27 (22%). ECMO was provided for one patient who survived.

Electrocardiographic abnormalities were present in 19/55 (35%) cases. Of these, 8 (42%) had atrial fibrillation, 7 (37%) had repolarisation abnormalities, 2 (11%) had conduction defects and 2 (11%) had supraventricular tachycardia. An echocardiogram was performed in 19/55 (35%) patients. Of these, left ventricular dysfunction was apparent in 5 (26%), 4 (21%) had a pericardial effusion, pulmonary hypertension was present in 3 (16%) and 1 patient had evidence of right ventricular infarction.

Patients, in general, received cautious intravenous therapy with simple, unbalanced crystalloid. The median (IQR) fluid balance over the initial 3 days of the ICU admission was only +1493 (175–3567) ml; only 11/55 (20%) were prescribed frusemide. In the 34/55 (62%) patients requiring vasopressor support, noradrenaline monotherapy was used in 18 (53%), adrenaline monotherapy was used in 2 (6%), while 14 (41%) required a combination of noradrenaline and adrenaline or vasopressin. Myocardial involvement with ventricular dysfunction was present in 6/34 (18%) of those requiring vasopressor support, however noradrenaline remained the predominant inotrope administered. No patients received dobutamine.

All patients received at least one antibiotic effective against leptospirosis, in 45/55 (82%) this was within the first 6 hours of presentation to hospital. A beta-lactam was the initial antibiotic in 42/55 (76%), doxycycline was prescribed in 9/55 (16%), while 4/55 (7%) received a macrolide. Corticosteroids were administered in 20/55 (36%) including the 2 patients that died. Intravenous hydrocortisone was the corticosteroid used in 17/20 (85%) patients; the

**Table 6. Characteristics of patients with severe leptospirosis requiring ventilatory support and intubation in Far North Queensland, Australia between 1997 and 2018.**

| Characteristic | Median (IQR) |
|---|---|
| Intubated at regional hospital (n (%)) | 7 (26%) |
| Intubated at tertiary hospital (n (%)) | 20 (74%) |
| Onset of symptoms until diagnosis of ARDS (days) | 5.5 (4.5–6.2) |
| First hospital presentation to intubation (hours) | 12.5 (8.4–31.5) |
| Time from ICU admission to intubation (hours) | 1.0 (-2.3 to 10.3) |
| PaO$_2$/FiO$_2$ ratio | 144 (82–205) |
| Highest positive end-expiratory pressure (cm H$_2$O) | 15 (10–15) |
| Peak inspiratory pressure at time of worst PaO$_2$/FiO$_2$ ratio (cm H$_2$O) | 30 (25–39) |
| Worst Murray score | 3.3 (2.8–3.5) |
| Tracheal intubation duration (days) | 8.6 (5.0–11.4) |

IQR: Interquartile range, ARDS: Acute Respiratory Distress Syndrome

**Table 7. Patient characteristic and interventions in patients with severe leptospirosis in relation to administration of corticosteroids in Far North Queensland, Australia between 1997 and 2018.**

| Patient characteristic and interventions | Corticosteroid administered n = 20 (%) | No corticosteroid administered n = 35 (%) | p |
|---|---|---|---|
| Age (IQR) | 47 (36–72) | 47 (27–59) | 0.12 |
| Gender (male) | 17 (85%) | 32 (91%) | 0.66 |
| Any comorbidity | 6 (30%) | 3 (9%) | 0.06 |
| APACHE III (IQR) | 96 (79–112) | 55 (42–85) | 0.005 |
| RRT | 10 (50%) | 8 (23%) | 0.07 |
| Vasopressor support | 18 (90%) | 16 (46%) | 0.001 |
| ARDS | 15 (75%) | 26 (74%) | 0.95 |
| Pulmonary haemorrhage | 13 (65%) | 19 (54%) | 0.44 |
| Died | 2 (10%) | 0 | 0.13 |

IQR: Interquartile range; APACHE: Acute Physiology And Chronic Health Evaluation; RRT: Renal replacement therapy; ARDS: Acute respiratory distress syndrome.

median (IQR) daily dose was 200 mg (150–300). Patients receiving corticosteroids had a higher APACHE III score than those that did not (median (IQR): 96 (79–112) versus 55 (42–85), p = 0.005). The characteristics of the patients who received corticosteroids are compared to those who did not in Table 7.

During the study period, in addition to the above management, all patients received early enteral nutritional support, stress ulcer prophylaxis and deep vein thrombosis prophylaxis where this was not contraindicated [24].

## Discussion

The early, comprehensive ICU support that can be provided in Australia's well-resourced public health system results in a case-fatality rate for leptospirosis which is among the lowest in the world. The patients in this cohort were critically ill and yet only two (3.6%) died, both of whom were elderly men with multiple medical comorbidities.

RRT was initiated using conventional indications. This was most commonly severe metabolic acidosis–the median arterial pH at the time of RRT initiation was 7.14 and the median base deficit was 14 –while almost a quarter had hypoxic respiratory failure, refractory fluid overload and anuric renal failure. In an ICU series from Réunion Island, the case-fatality rate of leptospirosis was reduced to 6% through the early initiation of organ support [10]. In that series, 95% had AKI and 56% received CVVHF, the majority within 24 hours of ICU admission. At the time of RRT initiation, the median urea was 25 mmol/L, the median pH was 7.38, the median base deficit of 3 mmol/L, and the median potassium was 3.7 mmol/L. Based on these results the study's authors recommended very early CVVHF, with initiation if two of the following criteria were met: urea >20 mmol/L; serum potassium >4 mmol/L; oligo-anuria without hypovolemia ascertained by echocardiography or hemodynamic monitoring devices [10]. However, our series confirms that while RRT has a crucial role in the management of these patients, waiting until more traditional thresholds are satisfied can still be associated with excellent outcomes. Other studies have also failed to identify any advantage in initiating RRT before conventional criteria are satisfied [25, 26]. Data supporting a more conservative approach to RRT initiation may provide reassurance to resource-poor low and middle-income countries (LMIC) where access to RRT is usually limited, particularly in the setting of outbreaks.

In the patients requiring RRT in our series, a continuous modality (CVVHF, continuous veno-venous haemodialysis or continuous veno-venous haemodiafiltration) was used in all

cases. In Brazil, daily haemodialysis was shown to improve mortality compared with delayed, alternate-day dialysis [9]. While prompt RRT is an essential component of the care of severe leptospirosis, the exact modality of RRT appears less important [27]. Furthermore, in settings where there is no access to haemodialysis, peritoneal dialysis has been associated with excellent outcomes in patients with leptospirosis-induced AKI, even in the setting of thrombocytopaenia [28, 29].

Pulmonary involvement in leptospirosis is due to impaired fluid handling of alveolar epithelial cells, leading to pulmonary oedema or–in the case of pulmonary haemorrhage–results from leptospiral proteins or toxic cellular components having a direct effect on the alveolocapillary membrane [30–32]. However, while different countries' guidelines provide criteria for the initiation of mechanical ventilation, the decision to intubate and ventilate a patient is almost always multifactorial. In Sri Lanka, bilateral infiltrates on chest X-ray plus a $PaO_2/FiO_2$ (PF ratio) $<200$ is considered an indication for mechanical ventilation, whereas in the Philippines, a PF ratio of $<250$ is used [3, 5]. In Fiji, non-invasive ventilation with CPAP or BIPAP is preferred initially, with intubation recommended if there is no improvement in oxygen saturation, or if PF ratio is $<100$ [4]. Lung protective ventilation and low tidal volumes are universally recommended, with an initial tidal volume of 6ml/kg and positive end expiratory pressure (PEEP) of 5 cmH$_2$O. Where plateau pressure can be measured, it should be maintained at $\leq 30$ cmH$_2$O [4, 5].

In our series, pulmonary involvement–with either pulmonary haemorrhage or ARDS–was common, occurring in 76% of cases. Non-invasive ventilation was used in almost one third of patients, however the majority received this for only a short period of time before deteriorating and requiring intubation and mechanical ventilation. Therefore, non-invasive ventilation can be considered in patients with severe leptospirosis, with the caveat that it is likely to represent only a temporising measure until the patient can be intubated and mechanically ventilated. Although many patients in our cohort had PF ratios consistent with severe ARDS, no particular criteria were used to inform the decision to intubate. This was instead based on the individual judgement of attending clinicians and informed by the presence of hypoxia or hypercapnoea, the respiratory rate, the work of breathing, alterations in mental state or more commonly, a combination of these factors. Therefore, while the patient's PF ratio can inform the decision to mechanically ventilate a patient, clinicians should avoid placing undue emphasis on this single value. Furthermore, a sudden event such as pulmonary haemorrhage or cardiac decompensation may necessitate more urgent intubation. For patients with severe hypoxic respiratory failure, prone ventilation and prostacyclin may be useful adjuncts to improve outcome, although the small number of patients receiving either intervention in this series precludes definitive recommendations. There is a paucity of evidence on the use of ECMO in severe leptospirosis, however it has been used successfully in patients who remain hypoxic despite maximal mechanical ventilation [33, 34]. In our series, ECMO was provided for one patient–a 37-year-old man with pulmonary haemorrhage resulting in severe hypoxia despite maximal ventilation, prone positioning, sedation and paralysis; he survived.

In the Philippines, it is recommended that hypovolaemic patients with leptospirosis and oliguric renal failure receive intravenous fluids until a urine output of 0.5ml/kg/hr is achieved. If this approach is unsuccessful, intravenous diuretics are added prior to consideration of RRT [5]. However, although 64% of our patients had oliguric renal failure and 62% required vasopressor support, intravenous fluids were administered cautiously. This reflects local clinicians' concern that liberal intravenous fluids will increase the risk of respiratory deterioration, particularly as concomitant lung involvement is common in cases of leptospirosis with renal failure and hypotension [7, 13]. Cautious fluid resuscitation is recommended in other tropical infections which can mimic leptospirosis, including rickettsial diseases and malaria [35, 36].

Furthermore, in LMIC, where most cases of leptospirosis, rickettsial diseases and malaria occur, access to mechanical ventilation is frequently limited [37]. Indeed, even where it is available, the patient who develops acute lung injury can still deteriorate rapidly and unpredictably [38]. Conversely, although hypovolaemia has the potential to exacerbate AKI, renal dysfunction develops less abruptly and is more easily remedied than pulmonary involvement [7, 9].

As with other infections, multiple organ dysfunction syndrome in leptospirosis carries a poor prognosis [2]. However, given the unique pathophysiology of leptospirosis infection and the high rate of cardiac involvement, the optimal approach to inotrope support is poorly defined. Dobutamine has been recommended in cases of severe leptospirosis, particularly in the context of myocarditis [3]. Although cardiac involvement was common in our series– 11% had evidence of ventricular dysfunction–noradrenaline was the predominant inotropic agent used, frequently with either adrenaline or vasopressin. These data suggest that the cause of shock in severe leptospirosis is most commonly distributive or hypovolaemic rather than cardiogenic. Cardiac involvement in leptospirosis, particularly myocarditis, is well described, with endocardial inflammation and vasculitis common autopsy findings [39, 40]. However, in critically ill patients, the cardiac manifestations of leptospirosis–including both left and right ventricular systolic dysfunction–are comparable to those seen in sepsis, providing support for the suggestion that the management should be similar [41].

The role of corticosteroids in leptospirosis is controversial with both benefit and harm being reported previously [15–17]. The proposed pathogenesis of leptospirosis induced lung injury includes a toxin-mediated vasculitis, an exaggerated host immune response and/or a non-cardiogenic pulmonary oedema triggered by a reduction in expression of the ENaC transporter in the luminal membrane of alveolar epithelial cells–all of which may be improved by corticosteroids [17, 42, 43]. In our series, over a third of patients received corticosteroids, including both the patients that died. The decision to initiate corticosteroid therapy was once again an individual clinical judgment rather than being based on any specific criteria. The fact that corticosteroids were administered in patients with higher APACHE III scores possibly reflects the influence of the sepsis literature that has supported the use of corticosteroids to hasten the resolution of shock [44], or perhaps–more likely–it is an indication of clinicians' limited therapeutic armamentarium for the most critical patients. Given the retrospective nature of the study, the small sample size and the fact that corticosteroids were delivered as only one component of a bundle of care, it was not possible to say that patients benefitted from this adjunctive treatment. A prospective randomised, controlled trial would be required to determine the role for corticosteroids in the management of leptospirosis and the optimal dose and duration of such therapy.

The heterogeneous approach to the management of severe leptospirosis around the world represents the limited evidence base, a near absence of multinational studies and a significant variation in access to sophisticated ICU support. The geographical variation in the predominant serovars may also be important. It is notable that there were no cases caused by serovar Icterohaemorrhagiae in our series which may explain the relatively infrequent finding of jaundice, hyperbilirubinaemia and severe thrombocytopenia in our patients. From a serological perspective, serovar Australis and serovar Zanoni accounted for most of the severe disease in our series. Serovar Icterohaemorrhagiae has been associated with severe disease in other series and may partly explain the high case-fatality rates seen in these reports. However, SOFA, SAPS II and APACHE III scores were high in our series and comparable to other series of severe disease [10, 45, 46]. Similarly, the clinical manifestations with the gravest prognosis–ARDS, pulmonary haemorrhage, renal involvement and hypotension–were also as common, suggesting that our findings may have generalisability.

While knowledge of the infecting serovar is unlikely to affect clinical management (and it is almost never available promptly enough to do so), it may influence the development of leptospirosis vaccines in humans, which thus far have been limited by poor efficacy, short protection periods and safety concerns [47].

This series emphasises the potential lethality of leptospirosis even in young individuals who are otherwise healthy. However, with prompt recognition of the infection and early provision of optimal supportive care, the prognosis appears good. Older age has previously been identified as a factor which increases the risk of leptospirosis-attributable mortality [2, 48, 49]; in this series, the 2 patients who died were elderly, both had multiple comorbidities, and both died with 48 hours of presentation to hospital suggesting that even optimal ICU support may have been ineffective.

Early recognition of severe disease is essential so that supportive care can be expedited. However, this is challenging, particularly as the microbiological diagnosis of leptospirosis can take several days even in well-resourced settings. The development of reliable point of care tests and validation of severity scoring tools, such as the SPiRO score–that are applicable in resource-poor settings–would not only assist in the early diagnosis of severe leptospirosis, but also identify those patients at greatest risk for deterioration, expediting their referral for supportive care [6, 50].

Our study has several limitations. Its retrospective nature and lack of standardised management made drawing more definitive recommendations difficult. There was, in particular, significant variation in the choice of antibiotic regimens and both the timing and dosing of corticosteroid therapy which precluded meaningful analysis of their impact on survival. However, the authors hope that documentation of the therapeutic strategies associated with the very low case-fatality rate seen in the cohort might contribute to the limited evidence base that informs the management of these patients.

In conclusion, the provision of prompt multimodal ICU support to patients with severe leptospirosis in tropical Australia results in a case-fatality rate which is amongst the lowest in the world. This bundle of care includes early recognition, prompt antibiotic therapy, protective ventilation techniques, initiation of RRT using traditional criteria, conservative intravenous fluids, and, in some cases, corticosteroids. However, prospective studies are required to determine the relative contribution of each of these interventions to optimal patient outcomes. Different therapeutic strategies that employ these interventions also require validation in settings where resources may be more limited and where different serovars are responsible for the disease.

## Supporting information

**S1 Checklist. STROBE checklist.**
(PDF)

## Acknowledgments

The authors would like to thank the medical records staff at Cairns, Innisfail, Tully and Atherton hospitals for their assistance in retrieving medical charts.

## Author Contributions

**Conceptualization:** Simon Smith, Angus Carter, Brendan J. Kennedy, Josh Hanson.

**Data curation:** Simon Smith, Yu-Hsuan Liu, Alexis Dermedgoglou, Suzanne S. Poulgrain, Matthew P. Paavola, Tarryn L. Minto, Michael Luc.

**Formal analysis:** Simon Smith, Josh Hanson.

**Methodology:** Simon Smith, Brendan J. Kennedy, Josh Hanson.

**Visualization:** Simon Smith.

**Writing – original draft:** Simon Smith, Josh Hanson.

**Writing – review & editing:** Simon Smith, Yu-Hsuan Liu, Angus Carter, Brendan J. Kennedy, Alexis Dermedgoglou, Suzanne S. Poulgrain, Matthew P. Paavola, Tarryn L. Minto, Michael Luc, Josh Hanson.

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
