## [Decision Letter · Decision Letter 0]

4 Nov 2019

Dear Dr Smith:

Thank you very much for submitting your manuscript "Severe leptospirosis in tropical Australia: optimising intensive care unit management to reduce mortality" (PNTD-D-19-01625) for review by PLOS Neglected Tropical Diseases. Your manuscript was fully evaluated at the editorial level and by independent peer reviewers. The reviewers appreciated the attention to an important topic but identified some aspects of the manuscript that should be improved.

We therefore ask you to modify the manuscript according to the review recommendations before we can consider your manuscript for acceptance. Your revisions should address the specific points made by each reviewer.

(1) A letter containing a detailed list of your responses to the review comments and a description of the changes you have made in the manuscript.

(2) Two versions of the manuscript: one with either highlights or tracked changes denoting where the text has been changed (uploaded as a "Revised Article with Changes Highlighted" file ); the other a clean version (uploaded as the article file).

(3) If available, a striking still image (a new image if one is available or an existing one from within your manuscript). If your manuscript is accepted for publication, this image may be featured on our website. Images should ideally be high resolution, eye-catching, single panel images; where one is available, please use 'add file' at the time of resubmission and select 'striking image' as the file type. 

Please provide a short caption, including credits, uploaded as a separate "Other" file. If your image is from someone other than yourself, please ensure that the artist has read and agreed to the terms and conditions of the Creative Commons Attribution License at http://journals.plos.org/plosntds/s/content-license (NOTE: we cannot publish copyrighted images). 

(4) Appropriate Figure Files 

Please remove all name and figure # text from your figure files upon submitting your revision. Please also take this time to check that your figures are of high resolution, which will improve both the editorial review process and help expedite your manuscript's publication should it be accepted. Please note that figures must have been originally created at 300dpi or higher. Do not manually increase the resolution of your files. For instructions on how to properly obtain high quality images, please review our Figure Guidelines, with examples at: http://journals.plos.org/plosntds/s/figures

While revising your submission, please upload your figure files to the Preflight Analysis and Conversion Engine (PACE) digital diagnostic tool, https://pacev2.apexcovantage.com/ PACE helps ensure that figures meet PLOS requirements. To use PACE, you must first register as a user. Then, login and navigate to the UPLOAD tab, where you will find detailed instructions on how to use the tool. If you encounter any issues or have any questions when using PACE, please email us at figures@plos.org.

We hope to receive your revised manuscript by Jan 03 2020 11:59PM. If you anticipate any delay in its return, we ask that you let us know the expected resubmission date by replying to this email.

To submit your revised files, please log in to https://www.editorialmanager.com/pntd/

Sincerely,

Joseph M. Vinetz

Deputy Editor

Melissa Caimano

Deputy Editor

Reviewer's Responses to Questions

**Key Review Criteria Required for Acceptance?**

**Methods**

-Are the objectives of the study clearly articulated with a clear testable hypothesis stated?

-Is the study design appropriate to address the stated objectives?

-Is the population clearly described and appropriate for the hypothesis being tested?

-Is the sample size sufficient to ensure adequate power to address the hypothesis being tested?

-Were correct statistical analysis used to support conclusions?

-Are there concerns about ethical or regulatory requirements being met?

Reviewer #1: In this retrospective study, 55 patients with leptospirosis admitted to Intensive Care Unity of one Hospital were evaluated. The methods are well described and the statistical analysis is adequated.

**Results**

-Does the analysis presented match the analysis plan?

-Are the results clearly and completely presented?

-Are the figures (Tables, Images) of sufficient quality for clarity?

Reviewer #1: The resulta are well presented. The number of Tables is sufficient.

**Conclusions**

-Are the conclusions supported by the data presented?

-Are the limitations of analysis clearly described?

-Do the authors discuss how these data can be helpful to advance our understanding of the topic under study?

-Is public health relevance addressed?

Reviewer #1: The conclusions were supported by the data presented. A prospective study is very important to confirm the findings of this study.

**Editorial and Data Presentation Modifications?**

Reviewer #1: Minor revision.

**Summary and General Comments**

Reviewer #1: The authors comment that corticosteroids therapy in these patients is indicated to toxin-mediated vasculitis or exaggerated host immune response (page 17, line 354). Experimental study demonstrated that in leptospirosis a non cardiogenic pulmonary edema may occur consequent to a reduction in expression of ENaC transporter in luminal membrane of the alveolar epithelial cells, since corticosteroids increases the expression of this transporter, it may be another mechanism that contributes to the treatment of ARDS in leptospirosis.

How many patients presented AKI with moderated to severe ARDS?

How many patients received furosemide?

Doxycycline was administered to patients with acute kidney injury?

PLOS authors have the option to publish the peer review history of their article (what does this mean?). If published, this will include your full peer review and any attached files.

Reviewer #1: No

---

## [Editor Report · Decision Letter 1]

15 Nov 2019

Dear Dr Smith,

We are pleased to inform you that your manuscript, "Severe leptospirosis in tropical Australia: optimising intensive care unit management to reduce mortality", has been editorially accepted for publication at PLOS Neglected Tropical Diseases.

Before your manuscript can be formally accepted and sent to production you will need to complete our formatting changes, which you will receive in a follow up email. Please note: your manuscript will not be scheduled for publication until you have made the required changes.

IMPORTANT NOTES

* Copyediting and Author Proofs: To ensure prompt publication, your manuscript will NOT be subject to detailed copyediting and you will NOT receive a typeset proof for review. The corresponding author will have one final opportunity to correct any errors when sent the requests mentioned above. Please review this version of your manuscript for any errors.

* If you or your institution will be preparing press materials for this manuscript, please inform our press team in advance at plosntds@plos.org. If you need to know your paper's publication date for media purposes, you must coordinate with our press team, and your manuscript will remain under a strict press embargo until the publication date and time. PLOS NTDs may choose to issue a press release for your article. If there is anything that the journal should know, please get in touch.

*Now that your manuscript has been provisionally accepted, please log into EM and update your profile. Go to http://www.editorialmanager.com/pntd, log in, and click on the "Update My Information" link at the top of the page. Please update your user information to ensure an efficient production and billing process.

*Note to LaTeX users only - Our staff will ask you to upload a TEX file in addition to the PDF before the paper can be sent to typesetting, so please carefully review our Latex Guidelines [http://www.plosntds.org/static/latexGuidelines.action] in the meantime.

Best regards,

Joseph M. Vinetz

Deputy Editor

Melissa Caimano

Deputy Editor

---

## [Editor Report · Acceptance letter]

21 Nov 2019

Dear Dr Smith,

We are delighted to inform you that your manuscript, "Severe leptospirosis in tropical Australia: optimising intensive care unit management to reduce mortality," has been formally accepted for publication in PLOS Neglected Tropical Diseases.

Best regards,

Serap Aksoy

Editor-in-Chief

Shaden Kamhawi

Editor-in-Chief
